# Efficient Algorithms for Non-convex Isotonic Regression through Submodular Optimization

**Francis Bach**
Inria
Département d'Informatique de l'Ecole Normale Supérieure
PSL Research University, Paris, France
`francis.bach@ens.fr`

## Abstract

We consider the minimization of submodular functions subject to ordering constraints. We show that this potentially non-convex optimization problem can be cast as a convex optimization problem on a space of uni-dimensional measures, with ordering constraints corresponding to first-order stochastic dominance. We propose new discretization schemes that lead to simple and efficient algorithms based on zero-th, first, or higher order oracles; these algorithms also lead to improvements without isotonic constraints. Finally, our experiments show that non-convex loss functions can be much more robust to outliers for isotonic regression, while still being solvable in polynomial time.

## 1 Introduction

Shape constraints such as ordering constraints appear everywhere in estimation problems in machine learning, signal processing and statistics. They typically correspond to prior knowledge, and are imposed for the interpretability of models, or to allow non-parametric estimation with improved convergence rates [16, 8]. In this paper, we focus on imposing ordering constraints into an estimation problem, a setting typically referred to as *isotonic* regression [4, 26, 22], and we aim to generalize the set of problems for which efficient (i.e., polynomial-time) algorithms exist.

We thus focus on the following optimization problem:

$$\min_{x \in [0,1]^n} H(x) \text{ such that } \forall (i,j) \in E, \ x_i \geqslant x_j, \tag{1}$$

where $E \subset \{1, \dots, n\}^2$ represents the set of constraints, which form a directed acyclic graph. For simplicity, we restrict $x$ to the set $[0,1]^n$, but our results extend to general products of (potentially unbounded) intervals.

As convex constraints, isotonic constraints are well-adapted to estimation problems formulated as convex optimization problems where $H$ is convex, such as for linear supervised learning problems, with many efficient algorithms for separable convex problems [4, 26, 22, 30], which can thus be used as inner loops in more general convex problems by using projected gradient methods (see, e.g., [3]).

In this paper, we show that another form of structure can be leveraged. We will assume that $H$ is *submodular*, which is equivalent, when twice continuously differentiable, to having nonpositive cross second-order derivatives. This notably includes *all* (potentially non convex) separable functions (i.e., sums of functions that depend on single variables), but also many other examples (see Section 2).

Minimizing submodular functions on continuous domains has been recently shown to be equivalent to a convex optimization problem on a space of uni-dimensional measures [2], and given that the functions $x \mapsto \lambda(x_j - x_i)_+$ are submodular for any $\lambda > 0$, it is natural that by using $\lambda$ tending to $+\infty$,

we recover as well a convex optimization problem; the main contribution of this paper is to provide a simple framework based on stochastic dominance, for which we design efficient algorithms which are based on simple oracles on the function $H$ (typically access to function values and derivatives). In order to obtain such algorithms, we go significantly beyond [2] by introducing novel discretization algorithms that also provide improvements without any isotonic constraints.

More precisely, we make the following contributions:

- We show in Section 3 that minimizing a submodular function with isotonic constraints can be cast as a convex optimization problem on a space of uni-dimensional measures, with isotonic constraints corresponding to first-order stochastic dominance.

- On top of the naive discretization schemes presented in Section 4, we propose in Section 5 new discretization schemes that lead to simple and efficient algorithms based on zero-th, first, or higher order oracles. They go from requiring $O(1/\varepsilon^3) = O(1/\varepsilon^{2+1})$ function evaluations to reach a precision $\varepsilon$, to $O(1/\varepsilon^{5/2}) = O(1/\varepsilon^{2+1/2})$ and $O(1/\varepsilon^{7/3}) = O(1/\varepsilon^{2+1/3})$.

- Our experiments in Section 6 show that non-convex loss functions can be much more robust to outliers for isotonic regression.

## 2    Submodular Analysis in Continuous Domains

In this section, we review the framework of [2] that shows how to minimize submodular functions using convex optimization.

**Definition.**    Throughout this paper, we consider a *continuous function* $H : [0,1]^n \to \mathbb{R}$. The function $H$ is said to be *submodular* if and only if [21, 29]:

$$\forall (x,y) \in [0,1]^n \times [0,1]^n, \ H(x) + H(y) \geqslant H(\min\{x,y\}) + H(\max\{x,y\}), \qquad (2)$$

where the min and max operations are applied component-wise. If $H$ is continuously twice differentiable, then this is equivalent to $\frac{\partial^2 H}{\partial x_i \partial x_j}(x) \leqslant 0$ for any $i \neq j$ and $x \in [0,1]^n$ [29].

The cone of submodular functions on $[0,1]^n$ is invariant by marginal strictly increasing transformations, and includes all functions that depend on a single variable (which play the role of linear functions for convex functions), which we refer to as *separable* functions.

**Examples.**    The classical examples are: (a) any separable function, (b) convex functions of the difference of two components, (c) concave functions of a positive linear combination, (d) negative log densities of multivariate totally positive distributions [17]. See Section 6 for a concrete example.

**Extension on a space of measures.**    We consider the convex set $\mathcal{P}([0,1])$ of *Radon probability measures* [24] on $[0,1]$, which is the closure (for the weak topology) of the convex hull of all Dirac measures. In order to get an *extension*, we look for a function defined on the set of *products of probability measures* $\mu \in \mathcal{P}([0,1])^n$, such that if all $\mu_i$, $i = 1, \ldots, n$, are Dirac measures at points $x_i \in [0,1]$, then we have a function value equal to $H(x_1, \ldots, x_n)$. Note that $\mathcal{P}([0,1])^n$ is different from $\mathcal{P}([0,1]^n)$, which is the set of probability measures on $[0,1]^n$.

For a probability distribution $\mu_i \in \mathcal{P}([0,1])$ defined on $[0,1]$, we can define the (reversed) cumulative distribution function $F_{\mu_i} : [0,1] \to [0,1]$ as $F_{\mu_i}(x_i) = \mu_i([x_i,1])$. This is a non-increasing left-continuous function from $[0,1]$ to $[0,1]$, such that $F_{\mu_i}(0) = 1$ and $F_{\mu_i}(1) = \mu_i(\{1\})$. See illustrations in the left plot of Figure 1.

We can then define the "inverse" cumulative function from $[0,1]$ to $[0,1]$ as $F_{\mu_i}^{-1}(t_i) = \sup\{x_i \in [0,1], \ F_{\mu_i}(x_i) \geqslant t_i\}$. The function $F_{\mu_i}^{-1}$ is non-increasing and right-continuous, and such that $F_{\mu_i}^{-1}(1) = \min \operatorname{supp}(\mu_i)$ and $F_{\mu_i}^{-1}(0) = 1$. Moreover, we have $F_{\mu_i}(x_i) \geqslant t_i \Leftrightarrow F_{\mu_i}^{-1}(t_i) \geqslant x_i$.

The extension from $[0,1]^n$ to the set of product probability measures is obtained by considering a single threshold $t$ applied to all $n$ cumulative distribution functions, that is:

$$\forall \mu \in \mathcal{P}([0,1])^n, \ h(\mu_1, \ldots, \mu_n) = \int_0^1 H\big[F_{\mu_1}^{-1}(t), \ldots, F_{\mu_n}^{-1}(t)\big] dt. \qquad (3)$$

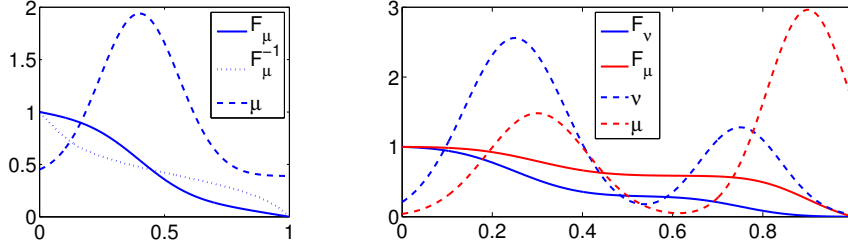

Figure 1: Left: cumulative and inverse cumulative distribution functions with the corresponding density (with respect to the Lebesgue measure). Right: cumulative functions for two distributions $\mu$ and $\nu$ such that $\mu \succcurlyeq \nu$.

We have the following properties when $H$ is submodular: (a) it is an extension, that is, if for all $i$, $\mu_i$ is a Dirac at $x_i$, then $h(\mu) = H(x)$; (b) it is convex; (c) minimizing $h$ on $\mathcal{P}([0,1])^n$ and minimizing $H$ on $[0,1]^n$ is equivalent; moreover, the minimal values are equal and $\mu$ is a minimizer if and only if $\left[F_{\mu_1}^{-1}(t), \dots, F_{\mu_n}^{-1}(t)\right]$ is a minimizer of $H$ for almost all $t \in [0,1]$. Thus, submodular minimization is equivalent to a convex optimization problem in a space of uni-dimensional measures.

Note that the extension is defined on all tuples of measures $\mu = (\mu_1, \dots, \mu_n)$ but it can equivalently be defined through non-increasing functions from $[0,1]$ to $[0,1]$, e.g., the representation in terms of cumulative distribution functions $F_{\mu_i}$ defined above (this representation will be used in Section 4 where algorithms based on the discretization of the equivalent obtained convex problem are discussed).

## 3 Isotonic Constraints and Stochastic Dominance

In this paper, we consider the following problem:

$$\inf_{x \in [0,1]^n} H(x) \text{ such that } \forall (i,j) \in E, x_i \geqslant x_j, \tag{4}$$

where $E$ is the edge set of a directed acyclic graph on $\{1, \dots, n\}$ and $H$ is submodular. We denote by $\mathcal{X} \subset \mathbb{R}^n$ (not necessarily a subset of $[0,1]^n$) the set of $x \in \mathbb{R}^n$ satisfying the isotonic constraints.

In order to define an extension in a space of measures, we consider a specific order on measures on $[0,1]$, namely *first-order stochastic dominance* [20], defined as follows.

Given two distributions $\mu$ and $\nu$ on $[0,1]$, with (inverse) cumulative distribution functions $F_\mu$ and $F_\nu$, we have $\mu \succcurlyeq \nu$, if and only if $\forall x \in [0,1]$, $F_\mu(x) \geqslant F_\nu(x)$, or equivalently, $\forall t \in [0,1]$, $F_\mu^{-1}(t) \geqslant F_\nu^{-1}(t)$. As shown in the right plot of Figure 1, the densities may still overlap. An equivalent characterization [19, 9] is the existence of a joint distribution on a vector $(X, X') \in \mathbb{R}^2$ with marginals $\mu(x)$ and $\nu(x')$ and such that $X \geqslant X'$ almost surely[1]. We now prove the main proposition of the paper:

**Proposition 1** *We consider the convex minimization problem:*

$$\inf_{\mu \in \mathcal{P}([0,1])^n} h(\mu) \text{ such that } \forall (i,j) \in E, \mu_i \succcurlyeq \mu_j. \tag{5}$$

*Problems in Eq. (4) and Eq. (5) have the same objective values. Moreover, $\mu$ is a minimizer of Eq. (5) if and only if $F_\mu^{-1}(t)$ is a minimizer of $H$ of Eq. (4) for almost all $t \in [0,1]$.*

**Proof** We denote by $\mathcal{M}$ the set of $\mu \in \mathbb{P}([0,1])^n$ satisfying the stochastic ordering constraints. For any $x \in [0,1]^n$ that satisfies the constraints in Eq. (4), i.e., $x \in \mathcal{X} \cap [0,1]^n$, the associated Dirac measures satisfy the constraint in Eq. (5). Therefore, the objective value $M$ of Eq. (4) is greater or equal to the one $M'$ of Eq. (5). Given a minimizer $\mu$ for the convex problem in Eq. (5), we have: $M \geqslant M' = h(\mu) = \int_0^1 H\left[F_{\mu_1}^{-1}(t), \dots, F_{\mu_n}^{-1}(t)\right] dt \geqslant \int_0^1 M\,dt = M$. This shows the proposition by studying the equality cases above. ∎

Alternatively, we could add the penalty term $\lambda \sum_{(i,j) \in \mathbb{E}} \int_{-\infty}^{+\infty} (F_{\mu_j}(z) - F_{\mu_i}(z))_+ dz$, which corresponds to the unconstrained minimization of $H(x) + \lambda \sum_{(i,j) \in \mathbb{E}} (x_j - x_i)_+$. For $\lambda > 0$ big enough[2], this is equivalent to the problem above, but with a submodular function which has a large Lipschitz constant (and is thus harder to optimize with the iterative methods presented below).

## 4 Discretization algorithms

Prop. 1 shows that the isotonic regression problem with a submodular cost can be cast as a convex optimization problem; however, this is achieved in a space of measures, which cannot be handled directly computationally in polynomial time. Following [2], we consider a polynomial time and space discretization scheme of each interval $[0,1]$ (and *not* of $[0,1]^n$), but we propose in Section 5 a significant improvement that allows to reduce the number of discrete points significantly. All pseudo-codes for the algorithms are available in Appendix B.

### 4.1 Review of submodular optimization in discrete domains

All our algorithms will end up minimizing approximately a submodular function $F$ on $\{0, \ldots, k-1\}^n$, that is, which satisfies Eq. (2). Isotonic constraints will be added in Section 4.2.

Following [2], this can be formulated as minimizing a convex function $f_\downarrow$ on the set of $\rho \in [0,1]^{n \times (k-1)}$ so that for each $i \in \{1, \ldots, n\}$, $(\rho_{ij})_{j \in \{1, \ldots, k-1\}}$ is a non-increasing sequence (we denote by $\mathcal{S}$ this set of constraints) corresponding to the cumulative distribution function. For any feasible $\rho$, a subgradient of $f_\downarrow$ may be computed by sorting all $n(k-1)$ elements of the matrix $\rho$ and computing at most $n(k-1)$ values of $F$. An approximate minimizer of $F$ (which exactly inherits approximation properties from the approximate optimality of $\rho$) is then obtained by selecting the minimum value of $F$ in the computation of the subgradient. Projected subgradient methods can then be used, and if $\Delta F$ is the largest absolute difference in values of $F$ when a single variable is changed by $\pm 1$, we obtain an $\varepsilon$-minimizer (for function values) after $t$ iterations, with $\varepsilon \leqslant nk\Delta F/\sqrt{t}$. The projection step is composed of $n$ simple separable quadratic isotonic regressions with chain constraints in dimension $k$, which can be solved easily in $O(nk)$ using the pool-adjacent-violator algorithm [4]. Computing a subgradient requires a sorting operation, which is thus $O(nk \log(nk))$. See more details in [2].

Alternatively, we can minimize the strongly-convex $f_\downarrow(\rho) + \frac{1}{2}\|\rho\|_F^2$ on the set of $\rho \in \mathbb{R}^{n \times (k-1)}$ so that for each $i$, $(\rho_{ij})_j$ is a non-increasing sequence, that is, $\rho \in \mathcal{S}$ (the constraints that $\rho_{ij} \in [0,1]$ are dropped). We then get a minimizer $z$ of $F$ by looking for all $i \in \{1, \ldots, n\}$ at the largest $j \in \{1, \ldots, k-1\}$ such that $\rho_{ij} \geqslant 0$. We take then $z_i = j$ (and if no such $j$ exists, $z_i = 0$). A gap of $\varepsilon$ in the problem above, leads to a gap of $\sqrt{\varepsilon nk}$ for the original problem (see more details in [2]). The subgradient method in the primal, or Frank-Wolfe algorithm in the dual may be used for this problem. We obtain an $\varepsilon$-minimizer (for function values) after $t$ iterations, with $\varepsilon \leqslant \Delta F/t$, which leads for the original submodular minimization problem to the same optimality guarantees as above, but with a faster algorithm in practice. See the detailed computations and comparisons in [2].

### 4.2 Naive discretization scheme

Following [2], we simply discretize $[0,1]$ by selecting the $k$ values $\frac{i}{k-1}$ or $\frac{2i+1}{2k}$, for $i \in \{0, \ldots, k-1\}$. If the function $H : [0,1]^n$ is $L_1$-Lipschitz-continuous with respect to the $\ell_1$-norm, that is $|H(x) - H(x')| \leqslant L_1\|x - x'\|_1$, the function $F$ is $(L_1/k)$-Lipschitz-continuous with respect to the $\ell_1$-norm (and thus we have $\Delta F \leqslant L_1/k$ above). Moreover, if $F$ is minimized up to $\varepsilon$, $H$ is optimized up to $\varepsilon + nL_1/k$.

In order to take into account the isotonic constraints, we simply minimize with respect to $\rho \in [0,1]^{n \times (k-1)} \cap \mathcal{S}$, with the additional constraint that for all $j \in \{1, \ldots, k-1\}$, $\forall (a,b) \in E$, $\rho_{a,j} \geqslant \rho_{b,j}$. This corresponds to additional contraints $\mathcal{T} \subset \mathbb{R}^{n \times (k-1)}$.

Following Section 4.1, we can either choose to solve the convex problem $\min_{\rho \in [0,1]^{n \times k} \cap S \cap T} f_\downarrow(\rho)$, or the strongly-convex problem $\min_{\rho \in S \cap T} f_\downarrow(\rho) + \frac{1}{2}\|\rho\|_F^2$. In the two situations, after $t$ iterations, that is $tnk$ accesses to values of $H$, we get a constrained minimizer of $H$ with approximation guarantee $nL_1/k + nL_1/\sqrt{t}$. Thus in order to get a precision $\varepsilon$, it suffices to select $k \geqslant 2nL_1/\varepsilon$ and $t \geqslant 4n^2 L_1^2/\varepsilon^2$, leading to an overall $8n^4 L_1^3/\varepsilon^3$ accesses to function values of $H$, which is the same as obtained in [2] (except for an extra factor of $n$ due to a different definition of $L_1$).

## 4.3 Improved behavior for smooth functions

We consider the discretization points $\frac{i}{k-1}$ for $i \in \{0, \ldots, k-1\}$, and we assume that all first-order (resp. second-order) partial derivatives are bounded by $L_1$ (resp. $L_2^2$). In the reasoning above, we may upper-bound the infimum of the discrete function in a finer way, going from $\inf_{x \in \mathcal{X}} H(x) + nL_1/k$ to $\inf_{x \in \mathcal{X}} H(x) + \frac{1}{2}n^2 L_2^2/k^2$ (by doing a Taylor expansion around the global optimum, where the first-order terms are always zero, either because the partial derivative is zero or the deviation is zero). We now select $k \geqslant nL_2/\sqrt{\varepsilon}$, leading to a number of accesses to $H$ that scales as $4n^4 L_1^2 L_2/\varepsilon^{5/2}$. We thus gain a factor $\sqrt{\varepsilon}$ with the exact same algorithm, but different assumptions.

## 4.4 Algorithms for isotonic problem

Compared to plain submodular minimization where we need to project onto $S$, we need to take into account the extra isotonic constraints, i.e., $\rho \in T$, and thus use more complex orthogonal projections.

**Orthogonal projections.** We now require the orthogonal projections on $S \cap T$ or $[0, 1]^{n \times k} \cap S \cap T$, which are themselves isotonic regression problems with $nk$ variables. If there are $m$ original isotonic constraints in Eq. (4), the number of isotonic constraints for the projection step is $O(nk + mk)$, which is typically $O(mk)$ if $m \geqslant n$, which we now assume. Thus, we can use existing parametric max-flow algorithms which can solve these in $O(nmk^2 \log(nk))$ [13] or in $O(nmk^2 \log(n^2k/m))$ [11]. See in Appendix A a description of the reformulation of isotonic regression as a parametric max-flow problem, and the link with minimum cut. Following [7, Prop. 5.3], we incorporate the $[0, 1]$ box constraints, by first ignoring them and thus by projecting onto the regular isotonic constraints, and then thresholding the result through $x \to \max\{\min\{x, 1\}, 0\}$.

Alternatively, we can explicitly consider a sequence of max-flow problems (with at most $\log(1/\varepsilon)$ of these, where $\varepsilon$ is the required precision) [28, 15]. Finally, we may consider (approximate) alternate projection algorithms such as Dykstra's algorithm and its accelerated variants [6], since the set $S$ is easy to project to, while, in some cases, such as chain isotonic constraints for the original problem, $T$ is easy to project to.

Finally, we could also use algorithms dedicated to special structures for isotonic regression (see [27]), in particular when our original set of isotonic constraints in Eq. (4) is a chain, and the orthogonal projection corresponds to a two-dimensional grid [26]. In our experiments, we use a standard max-flow code [5] and the usual divide-and-conquer algorithms [28, 15] for parametric max-flow.

**Separable problems.** The function $f_\downarrow$ from Section 4.2 is then a linear function of the form $f_\downarrow(\rho) = \operatorname{tr} w^\top \rho$, and then, a single max-flow algorithm can be used.

For these separable problberms, the alternative strongly-convex problem of minimizing $f_\downarrow(\rho) + \frac{1}{2}\|\rho\|_F^2$ becomes the one of minimizing $\min_{\rho \in S \cap T} \frac{1}{2}\|\rho + w\|_F^2$, which is simply the problem of projecting on the intersection of two convex sets, for which an accelerated Dykstra algorithm may be used [6], with convergence rate in $O(1/t^2)$ after $t$ iterations. Each step is $O(kn)$ for projecting onto $S$, while this is $k$ parametric network flows with $n$ variables and $m$ constraints for projecting onto $T$, in $O(knm \log n)$ for the general case and $O(kn)$ for chains and rooted trees [4, 30].

In our experiments in Section 6, we show that Dykstra's algorithm converges quickly for separable problems. Note that when the underlying losses are convex[3], then Dykstra converges in a *single* iteration. Indeed, in this situation, the sequences $(-w_{ij})_j$ are non-increasing and isotonic regression

along a direction preserves decreasingness in the other direction, which implies that after two alternate projections, the algorithm has converged to the optimal solution.

Alternatively, for the non-strongly convex formulation, this is a single network flow problem with $n(k-1)$ nodes, and $mk$ constraints, in thus $O(nmk^2 \log(nk))$ [25]. When $E$ corresponds to a chain, then this is a 2-dimensional-grid with an algorithm in $O(n^2k^2)$ [26]. For a precision $\varepsilon$, and thus $k$ proportional to $n/\varepsilon$ with the assumptions of Section 4.2, this makes a number of function calls for $H$, equal to $O(kn) = O(n^2/\varepsilon)$ and a running-time complexity of $O(n^3m/\varepsilon^2 \cdot \log(n^2/\varepsilon))$—for smooth functions, as shown in Section 4.3, we get $k$ proportional to $n/\sqrt{\varepsilon}$ and thus an improved behavior.

# 5 Improved discretization algorithms

We now consider a different discretization scheme that can take advantage of access to higher-order derivatives. We divide $[0, 1]$ into $k$ disjoint pieces $A_0 = [0, \frac{1}{k})$, $A_1 = [\frac{1}{k}, \frac{2}{k})$, ..., $A_{k-1} = [\frac{k-1}{k}, 1]$. This defines a new function $\tilde{H} : \{0, \ldots, k-1\}^n \to \mathbb{R}$ defined *only* for elements $z \in \{0, \ldots, k-1\}^n$ that satisfy the isotonic constraint, i.e., $z \in \{0, \ldots, k-1\}^n \cap \mathcal{X}$:

$$\tilde{H}(z) = \min_{x \in \prod_{i=1}^n A_{z_i}} H(x) \text{ such that } \forall (i, j) \in E, x_i \geqslant x_j. \tag{6}$$

The function $\tilde{H}(z)$ is equal to $+\infty$ if $z$ does not satisfy the isotonic constraints.

**Proposition 2** *The function $\tilde{H}$ is submodular, and minimizing $\tilde{H}(z)$ for $z \in \{0, \ldots, k-1\}^n$ such that $\forall (i, j) \in E, z_i \geqslant z_j$ is equivalent to minimizing Eq. (4).*

**Proof** We consider $z$ and $z'$ that satisfy the isotonic constraints, with minimizers $x$ and $x'$ in the definition in Eq. (6). We have $H(z) + H(z') = H(x) + H(x') \geqslant H(\min\{x, x'\}) + H(\max\{x, x'\}) \geqslant H(\min\{z, z'\}) + H(\max\{z, z'\})$. Thus it is submodular on the sub-lattice $\{0, \ldots, k-1\}^n \cap \mathcal{X}$. ∎

Note that in order to minimize $\tilde{H}$, we need to make sure that we only access $H$ for elements $z$ that satisfy the isotonic constraints, that is $\rho \in \mathcal{S} \cap \mathcal{T}$ (which our algorithms impose).

## 5.1 Approximation from high-order smoothness

The main idea behind our discretization scheme is to use high-order smoothness to approximate for any required $z$, the function value $\tilde{H}(z)$. If we assume that $H$ is $q$-times differentiable, with uniform bounds $L_r^r$ on all $r$-th order derivatives, then, the $(q-1)$-th order Taylor expansion of $H$ around $y$ is equal to $H_q(x|y) = H(y) + \sum_{r=1}^{q-1} \sum_{|\alpha|=r} \frac{1}{\alpha!}(x - y)^\alpha H^{(\alpha)}(y)$, where $\alpha \in \mathbb{N}^n$ and $|\alpha|$ is the sum of elements, $(x - y)^\alpha$ is the vector with components $(x_i - y_i)^{\alpha_i}$, $\alpha!$ the products of all factorials of elements of $\alpha$, and $H^{(\alpha)}(y)$ is the partial derivative of $H$ with order $\alpha_i$ for each $i$.

We thus approximate $\tilde{H}(z)$, for any $z$ that satisfies the isotonic constraint (i.e., $z \in \mathcal{X}$), by $\hat{H}(z) = \min_{x \in (\prod_{i=1}^n A_{z_i}) \cap \mathcal{X}} H_q(x|\frac{z+1/2}{k})$. We have for any $z$, $|\tilde{H}(z) - \hat{H}(z)| \leqslant (nL_q/2k)^q/q!$. Moreover, when moving a single element of $z$ by one, the maximal deviation is $L_1/k + 2(nL_q/2k)^q/q!$.

If $\hat{H}$ is submodular, then the same reasoning as in Section 4.2 leads to an approximate error of $(nk/\sqrt{t})(L_1/k + 2(nL_q/2k)^q/q!)$ after $t$ iterations, on top of $(nL_q/2k)^q/q!$, thus, with $t \geqslant 16n^2L_1^2/\varepsilon^2$ and $k \geqslant (q!\varepsilon/2)^{-1/q}nL_q/2$ (assuming $\varepsilon$ small enough such that $t \geqslant 16n^2k^2$), this leads to a number of accesses to the $(q-1)$-th order oracle equal to $O(n^4L_1^2L_q/\varepsilon^{2+1/q})$. We thus get an improvement in the power of $\varepsilon$, which tend to $\varepsilon^{-2}$ for infinitely smooth problems. Note that when $q = 1$ we recover the same rate as in Section 4.3 (with the same assumptions but a slightly different algorithm).

However, unless $q = 1$, the function $\hat{H}(z)$ is not submodular, and we cannot apply directly the bounds for convex optimization of the extension. We show in Appendix D that the bound still holds for $q > 1$ by using the special structure of the convex problem.

What remains unknown is the computation of $\hat{H}$ which requires to minimize polynomials on a small cube. We can always use the generic algorithms from Section 4.2 for this, which do not access extra

function values but can be slow. For quadratic functions, we can use a convex relaxation which is not tight but already allows strong improvements with much faster local steps, and which we now present. See the pseudo-code in Appendix B. In any case, using expansions of higher order is only practically useful in situations where function evaluations are expensive.

## 5.2 Quadratic problems

In this section, we consider the minimization of a quadratic submodular function $H(x) = \frac{1}{2}x^\top A x + c^\top x$ (thus with all off-diagonal elements of $A$ non-negative) on $[0,1]^n$, subject to isotonic constraints $x_i \geqslant x_j$ for all $(i,j) \in E$. This is the sub-problem required in Section 5.1 when using second-order Taylor expansions.

It could be solved iteratively (and approximately) with the algorithm from Section 4.2; in this section, we consider a semidefinite relaxation which is tight for certain problems ($A$ positive semi-definite, $c$ non-positive, or $A$ with non-positive diagonal elements), but not in general (we have found counter-examples but it is most often tight).

The relaxation is based on considering the set of $(Y,y) \in \mathbb{R}^{n\times n} \times \mathbb{R}^n$ such that there exists $x \in [0,1]^n \cap \mathfrak{X}$ with $Y = xx^\top$ and $y = x$. Our problem is thus equivalent to minimizing $\frac{1}{2}\operatorname{tr} AY + c^\top y$ such that $(Y,y)$ is in the convex-hull $\mathcal{Y}$ of this set, which is NP-hard to characterize in polynomial time [10]. However, following ideas from [18], we can find a simple relaxation by considering the following constraints: (a) for all $i \neq j$, $\begin{pmatrix} Y_{ii} & Y_{ij} & y_i \\ Y_{ij} & Y_{jj} & y_j \\ y_i & y_j & 1 \end{pmatrix}$ is positive semi-definite, (b) for all $i \neq j$, $Y_{ij} \leqslant \inf\{y_i, y_j\}$, which corresponds to $x_i x_j \leqslant \inf\{x_i, x_j\}$ for any $x \in [0,1]^n$, (c) for all $i$, $Y_{ii} \leqslant y_i$, which corresponds to $x_i^2 \leqslant x_i$, and (d) for all $(i,j) \in E$, $y_i \geqslant y_j$, $Y_{ii} \geqslant Y_{jj}$, $Y_{ij} \geqslant \max\{Y_{jj}, y_j - y_i + Y_{ii}\}$ and $Y_{ij} \leqslant \max\{Y_{ii}, y_i - y_j + Y_{jj}\}$, which corresponds to $x_i \geqslant x_j$, $x_i^2 \geqslant x_j^2$, $x_i x_j \geqslant x_j^2$, $x_i(1-x_i) \leqslant x_i(1-x_j)$, $x_i x_j \leqslant x_i^2$, and $x_i(1-x_j) \geqslant x_j(1-x_j)$. This leads to a semi-definite program which provides a lower-bound on the optimal value of the problem. See Appendix E for a proof of tightness for special cases and a counter-example for the tightness in general.

# 6 Experiments

We consider experiments aiming at (a) showing that the new possibility of minimizing submodular functions with isotonic constraints brings new possibilities and (b) that the new discretization algorithms are faster than the naive one.

**Robust isotonic regression.** Given some $z \in \mathbb{R}^n$, we consider a separable function $H(x) = \frac{1}{n}\sum_{i=1}^n G(x_i - z_i)$ with various possibilities for $G$: (a) the square loss $G(t) = \frac{1}{2}t^2$, (b) the absolute loss $G(t) = |t|$ and (c) a logarithmic loss $G(t) = \frac{\kappa^2}{2}\log\left(1 + t^2/\kappa^2\right)$, which is the negative log-density of a Student distribution and non-convex. The non-convexity of the cost function and the fact that is has vanishing derivatives for large values make it a good candidate for robust estimation [12]. The first two losses may be dealt with methods for separable convex isotonic regression [22, 30], but the non-convex loss can only dealt with exactly by the new optimization routine that we present—majorization-minimization algorithms [14] based on the concavity of $G$ as a function of $t^2$ can be used with such non-convex losses, but as shown below, they converge to bad local optima.

For simplicity, we consider chain constraints $1 \geqslant x_1 \geqslant x_2 \geqslant \cdots \geqslant x_n \geqslant 0$. We consider two set-ups: (a) a separable set-up where maximum flow algorithms can be used directly (with $n = 200$), and (b) a general submodular set-up (with $n = 25$ and $n = 200$), where we add a smoothness penalty which is the sum of terms of the form $\frac{\lambda}{2}\sum_{i=1}^{n-1}(x_i - x_{i+1})^2$, which is submodular (but not separable).

**Data generation.** We generate the data $z \in \mathbb{R}^n$, with $n = 200$, as follows: we first generate a simple *decreasing* function of $i \in \{1, \ldots, n\}$ (here an affine function); we then perturb this ground truth by (a) adding some independent noise and (b) corrupting the data by changing a random subset of the $n$ values by the application of another function which is *increasing* (see Figure 2, left). This is an adversarial perturbation, while the independent noise is not adversarial; the presence of the adversarial noise makes the problem harder as the proportion of corrupted data increases.

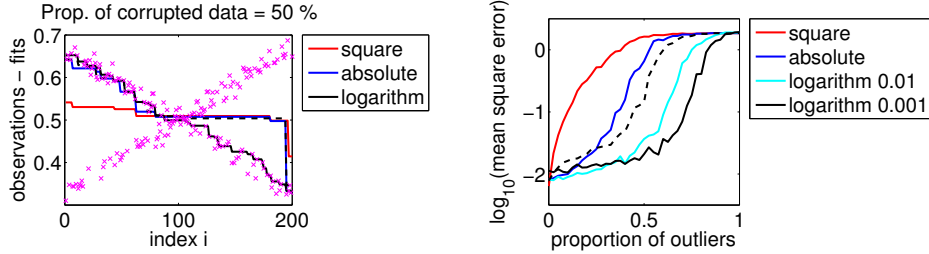

Figure 2: Left: robust isotonic regression with decreasing constraints, with $50\%$ of corrupted data (observation in pink crosses, and results of isotonic regression with various losses in red, blue and black); the dashed black line corresponds to majorization-minimization algorithm started from the observations. Right: robustness of various losses to the proportion of corrupted data. The two logarithm-based losses are used with two values of $\kappa$ (0.01 and 0.001); the dashed line corresponds to the majorization-minimization algorithm (with no convergence guarantees and worse performance).

**Optimization of separable problems with maximum flow algorithms.** We solve the discretized version by a single maximum-flow problem of size $nk$. We compare the various losses for $k = 1000$ on data which is along a decreasing line (plus noise), but corrupted (i.e., replaced for a certain proportion) by data along an increasing line. See an example in the left plot of Figure 2 for $50\%$ of corrupted data. We see that the square loss is highly non robust, while the (still convex) absolute loss is slightly more robust, and the robust non-convex loss still approximates the decreasing function correctly with $50\%$ of corrupted data when optimized globally, while the method with no guarantee (based on majorization-minimization, dashed line) does not converge to an acceptable solution. In Appendix C, we show additional examples where it is robust up to $75\%$ of corruption.

In the right plot of Figure 2, we also show the robustness to an increasing proportion of outliers (for the same type of data as for the left plot), by plotting the mean-squared error in log-scale and averaged over 20 replications. Overall, this shows the benefits of non-convex isotonic regression with guaranteed global optimization, even for large proportions of corrupted data.

**Optimization of separable problems with pool-adjacent violator (PAV) algorithm.** As shown in Section 4.2, discretized separable submodular optimization corresponds to the orthogonal projection of a matrix into the intersection of chain isotonic constraints in each row, and isotonic constraints in each column equal to the original set of isotonic constraints (in these simulations, these are also chain constraints). This can be done by Dykstra's alternating projection algorithm or its accelerated version [6], for which each projection step can be performed with the PAV algorithm because each of them corresponds to chain constraints.

In the left plot of Figure 3, we show the difference in function values (in log-scale) for various discretization levels (defined by the integer $k$ spaced by $1/4$ in base-10 logarithm), as as function of the number of iterations (averaged over 20 replications). For large $k$ (small difference of function values), we see a spacing between the ends of the plots of approximatively $1/2$, highlighting the dependence in $1/k^2$ of the final error with discretization $k$, which our analysis in Section 4.3 suggests.

**Effect of the discretization for separable problems.** In order to highlight the effect of discretization and its interplay with differentiability properties of the function to minimize, we consider in the middle plot of Figure 3, the distance in function values after full optimization of the discrete submodular function for various values of $k$. We see that for the simple smooth function (quadratic loss), we have a decay in $1/k^2$, while for the simple non smooth function (absolute loss), we have a final decay in $1/k$), a predicted by our analysis. For the logarithm-based loss, whose smoothness constant depends on $\kappa$, when $\kappa$ is large, it behaves like a smooth function immediately, while for $\kappa$ smaller, $k$ needs to be large enough to reach that behavior.

**Non-separable problems.** We consider adding a smoothness penalty to add the prior knowledge that values should be decreasing *and* close. In Appendix C, we show the effect of adding a smoothness prior (for $n = 200$): it leads to better estimation. In the right plot of Figure 3, we show the effect of various discretization schemes (for $n = 25$), from order 0 (naive discretization), to order 1 and 2

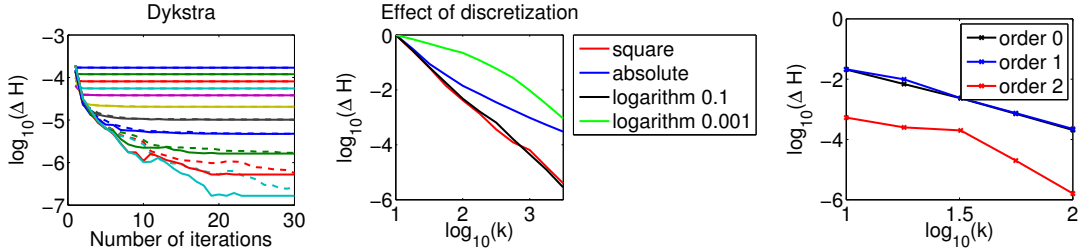

Figure 3: Dykstra's projection algorithms for separable problems, with several values of $k$, spaced with $1/4$ in base-10 logarithm, from $10^{-1}$ to $10^{-3.5}$. Dykstra in dashed and accelerated Dykstra in plain. Middle: effect of discretization value $k$ for various loss functions for separable problems (the logarithm-based loss is considered with two values of $\kappa$, $\kappa = 0.1$ and $\kappa = 0.001$). Right: effect of discretization $k$ on non-separable problems.

(our new schemes based on Taylor expansions from Section 5.1), and we plot the difference in function values after 50 steps of subgradient descent: in each plot, the quantity $\Delta H$ is equal to $H(x_k^*) - H^*$, where $x_k^*$ is an approximate minimizer of the discretized problem with $k$ values and $H^*$ the minimum of $H$ (taking into account the isotonic constraints). As outlined in our analysis, the first-order scheme does not help because our function has bounded Hessians, while the second-order does so significantly.

# 7    Conclusion

In this paper, we have shown how submodularity could be leveraged to obtain polynomial-time algorithms for isotonic regressions with a submodular cost, based on convex optimization in a space of measures—although based on convexity arguments, our algorithms apply to all separable *non-convex* functions. The final algorithms are based on discretization, with a new scheme that also provides improvements based on smoothness (also without isotonic constraints). Our framework is worth extending in the following directions: (a) we currently consider a fixed discretization, it would be advantageous to consider adaptive schemes, potentially improving the dependence on the number of variables $n$ and the precision $\varepsilon$; (b) other shape constraints can be consider in a similar submodular framework, such as $x_i x_j \geqslant 0$ for certain pairs $(i, j)$; (c) a direct convex formulation without discretization could probably be found for quadratic programming with submodular costs (which are potentially non-convex but solvable in polynomial time); (d) a statistical study of isotonic regression with adversarial corruption could now rely on formulations with polynomial-time algorithms.

**Acknowledgements**

We acknowledge support the European Research Council (grant SEQUOIA 724063).

## Footnotes

[1]Such a joint distribution may be built as the distribution of $(F_\mu^{-1}(T), F_\nu^{-1}(T))$, where $T$ is uniformly distributed in $[0,1]$.

[2]A short calculation shows that when $H$ is differentiable, the first order-optimality condition (which is only necessary here) implies that if $\lambda$ is strictly larger than $n$ times the largest possible partial first-order derivative of $H$, the isotonic constraints have to be satisfied.

[3]This is a situation where direct algorithms such as the ones by [22] are much more efficient than our discretization schemes.

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
