[Supplementary Material]

This is the supplementary material for the NIPS 2018 paper: "Efficient Algorithms for Non-convex Isotonic Regression through Submodular Optimization", by Francis Bach.

## A  Parametric max-flow formulation for isotonic regression

In this section, we provide descriptions of algorithms for isotonic regression with the quadratic cost, that is, the problem of solving:

$$\min_{x \in \mathbb{R}^n} \frac{1}{2} \|x - z\|_2^2 \text{ such that } \forall (i,j) \in E, \ x_i \geqslant x_j, \tag{7}$$

For more details, see [22, 1]. Note that in this section, we strongly leverage the submodular properties of cut functions.

The problem in Eq. (7) can be solved by considering the following penalized problem

$$\min_{x \in \mathbb{R}^n} \frac{1}{2} \|x - z\|_2^2 + \lambda \sum_{(i,j) \in E} (x_j - x_i)_+, \tag{8}$$

for a large value of $\lambda$. Note that as opposed to the general submodular case, having a large value does not impact running-time complexity because we are using combinatorial algorithms for min-cut / max-flow.

The function $g(x) = \sum_{(i,j) \in E} (x_j - x_i)_+$ is known to be the continuous *Lovász* extension of the cut function defined on $\{0,1\}^n$ with the exact same formula (see, e.g., [1]). Thus the problem in Eq. (8) is equivalent to

$$\min_{x \in \mathbb{R}^n} \frac{1}{2} \|x\|^2 - x^\top z + g(x), \tag{9}$$

which is the minimization of the Lovász extension $g(x)$ penalized by a separable function. This is known to be equivalent to the parameterized family of binary submodular function minimization problems

$$\min_{x \in \{0,1\}^n} \alpha \cdot 1_n^\top x - x^\top z + g(x), \tag{10}$$

for $\alpha \in \mathbb{R}$. More precisely (see, e.g., [1, Prop. 8.3]), one can obtain the unique solution $x \in \mathbb{R}^n$ of Eq. (9) from solutions $x_\alpha \in \{0,1\}^n$ of Eq. (10) through $x_i = \sup \big( \{\alpha \in \mathbb{R}, \ (x_\alpha)_i = 1\} \big)$.

Since the problem in Eq. (10) is a minimum cut problem, because of the equivalence with maximum flow we obtain a parametric max-flow problem.

## B  Pseudo-codes of algorithms

The paper proposes two discretization schemes: the naive one in Section 4.2 (which is the same as in [2] except with the added isotonic constraints) and the new improved one in Section 5.

### B.1  Naive discretization scheme

We consider the discretization of $[0,1]$ with $k$ elements $\frac{i}{k-1}$, $i \in \{0, \dots, k-1\}$. This defines a function $F$ on $\{0, \dots, k-1\}^n$ as

$$F(i_1, \dots, i_n) = H\Big(\frac{i_1}{k-1}, \dots, \frac{i_n}{k-1}\Big).$$

Following [2], we can define an extension in a space of product measures on $\{0, \dots, k-1\}$. We can parameterize that space through $\rho \in \mathbb{R}^{n \times (k-1)}$ which corresponds to the $n$ reverse cumulative distribution functions; minimizing $F$ (which leads to an approximate minimizer of $H$) is equivalent to minimizing a convex extension $f_\downarrow(\rho)$ with respect to $\rho \in [0,1]^n \cap \mathcal{S} \cap \mathcal{T}$ ($\mathcal{S}$ corresponds to the monotonicity of the cumulative distributions functions and $\mathcal{T}$ to the additional isotonic constraint). The only difference with the non-isotonic case is the extra-projection step onto $\mathcal{T}$.

**Projected subgradient method.** A subgradient of $f_\downarrow$ may be obtained by the greedy algorithm of [2], which sorts all values of the components of the matrix $\rho$, and computes differences of values of $F$ (and thus of $H$). The optimization algorithm is thus as follows (this is the subgradient method):

- **Initialization**: Choose discretization order $k$, maximum number of iterations $T$, step-size $\gamma$, and set $\rho_{ij}^0 = 1/2$ for all $(i,j)$
- **Subgradient iterations**: for $t = 1$ to $T$,
    - Compute a subgradient $w^{t-1} \in \mathbb{R}^{n \times (k-1)}$ of $f_\downarrow$ at $\rho^{t-1}$ using the greedy algorithm of [2]
    - Take a step $\tilde{\rho}^t = \rho^{t-1} - \frac{\gamma}{\sqrt{t}} w^{t-1}$
    - Compute the projection $\rho^t$ of $\tilde{\rho}^t$ onto $[0,1]^n \cap \mathcal{S} \cap \mathcal{T}$ using any parametric max-flow algorithm (this is an isotonic regression problem with quadratic cost)
- **Output**: $\left(\frac{i_1}{k-1}, \ldots, \frac{i_n}{k-1}\right)$ such that $(i_1, \ldots, i_n)$ leads to the maximal value of $F(i_1, \ldots, i_n)$ in the greedy algorithm for computing $w^T$.

**Strongly-convex subgradient method.** Here, we minimize $f_\downarrow(\rho) + \frac{1}{2}\|\rho\|_F^2$ with respect to $\rho \in \mathcal{S} \cap \mathcal{T}$, with the following subgradient method with an explicit step-size for strongly-convex problems (Frank-Wolfe could also be used for the dual problem):

- **Initialization**: Choose discretization order $k$, maximum number of iterations $T$, and set $\rho_{ij}^0 = 0$ for all $(i,j)$
- **Subgradient iterations**: for $t = 1$ to $T$,
    - Compute a subgradient $w^{t-1} \in \mathbb{R}^{n \times (k-1)}$ of $f_\downarrow$ at $\rho^{t-1}$ using the greedy algorithm of [2]
    - Take a step $\tilde{\rho}^t = (1 - \frac{2}{t+1})\rho^{t-1} - \frac{2}{t+1} w^{t-1}$
    - Compute the projection $\rho^t$ of $\tilde{\rho}^t$ onto $\mathcal{S} \cap \mathcal{T}$ using any parametric max-flow algorithm (this is an isotonic regression problem with quadratic cost)
- **Output**: $\left(\frac{i_1}{k-1}, \ldots, \frac{i_n}{k-1}\right)$ such that $(i_1, \ldots, i_n)$ leads to the maximal value of $F(i_1, \ldots, i_n)$ in the greedy algorithm for computing $w^T$.

### B.2 Improve discretization scheme

We are still here minimizing a function on $\{0, \ldots, k-1\}^n$, but this time we aim at minimizing with respect to $z \in \{0, \ldots, k-1\}^n \cap \mathcal{X}$ the submodular function

$$\tilde{H}(z) = \min_{x \in \prod_{i=1}^n A_{z_i}} H(x) \text{ such that } \forall (i,j) \in E, x_i \geqslant x_j,$$

where $A_0 = [0, \frac{1}{k})$, $A_1 = [\frac{1}{k}, \frac{2}{k})$, ..., $A_{k-1} = [\frac{k-1}{k}, 1]$. It can be approximated using a Taylor expansion by

$$\hat{H}(z) = \min_{x \in \prod_{i=1}^n A_{z_i}} H(\tfrac{z+1/2}{k}) + \sum_{r=1}^{q-1} \sum_{|\alpha|=r} \frac{1}{\alpha!} (x - \tfrac{z+1/2}{k})^\alpha H^{(\alpha)}(\tfrac{z+1/2}{k}) \text{ s. t. } \forall (i,j) \in E, x_i \geqslant x_j,$$

which only requires accesses to $H$ and its derivatives at $\frac{z+1/2}{k}$. This is trivial for $q = 1$, and can be approximated by a semidefinite program for $q = 2$ (see Section 5.2). We can then apply the subgradient method to $\hat{H}$ as follows:

- **Initialization**: Choose discretization order $k$, maximum number of iterations $T$, step-size $\gamma$ and set $\rho_{ij}^0 = 1/2$ for all $(i,j)$,
- **Subgradient iterations**: for $t = 1$ to $T$,
    - Compute a subgradient $w^{t-1} \in \mathbb{R}^{n \times (k-1)}$ of $\hat{h}_\downarrow$ at $\rho^{t-1}$ using the greedy algorithm of [2]
    - Take a step $\tilde{\rho}^t = \rho^{t-1} - \frac{\gamma}{\sqrt{t}} w^{t-1}$
    - Compute the projection $\rho^t$ of $\tilde{\rho}^t$ onto $[0,1]^n \cap \mathcal{S} \cap \mathcal{T}$ using any parametric max-flow algorithm (this is an isotonic regression problem with quadratic cost)
- **Output**: $\left(\frac{i_1+1/2}{k}, \ldots, \frac{i_n+1/2}{k}\right)$ such that $(i_1, \ldots, i_n)$ leads to the maximal value of $\hat{H}(i_1, \ldots, i_n)$ in the greedy algorithm for computing $w^T$.

## C  Additional experimental results

We first present here additional results, for robustness of isotonic regression to corrupted data in Figure 4, where we show that up to $75\%$ of corrupted data, the non-convex loss (solved exactly using submodularity) still finds a reasonable answer (but does not for $90\%$ of corruption). Then, in Figure 5, we present the effect of adding a smoothness term on top of isotonic constraints: we indeed get a smoother function as expected, and the higher-order algorithms perform significantly better.

## D  Approximate optimization for high-order discretization

In this section, we consider the set-up of Section 5, and we consider the minimization of the extension $\tilde{h}_\downarrow$ on $\mathcal{S} \cap \mathcal{T} \cap [0,1]^{n \times (k-1)}$. We consider the projected subgradient method, which uses an approximate subgradient not from $\tilde{h}_\downarrow$ but from the approximation $\hat{h}_\downarrow$, which we know is obtained from a function $\hat{H}$ such that $|\hat{H}(z) - \tilde{H}(z)| \leqslant \eta$ for all $z$, and for $\eta = (nL_q/2k)^q/q!$.

The main issue is that the extension $\hat{h}_\downarrow$ is not convex when $\hat{H}$ is not submodular (which could be the case because it is only an approximation of a submodular function). In order to show that the same projected subgradient method converges to an $\eta$-minimizer of $\tilde{h}_\downarrow$ (and hence of $\tilde{H}$), we simply consider a minimizer $\rho^*$ of $\tilde{h}_\downarrow$ (which is convex) on $\mathcal{S} \cap \mathcal{T} \cap [0,1]^{n \times (k-1)}$. Because of properties of submodular optimization problems, we may choose $\rho^*$ so that it takes values only in $\{0,1\}^{n \times (k-1)}$.

At iteration $t$, given $\rho^{t-1} \in \mathcal{S} \cap \mathcal{T} \cap [0,1]^{n \times (k-1)}$, we compute an approximate subgradient $\hat{w}^{t-1}$ using the greedy algorithm of [2] applied to $\hat{w}^{t-1}$. This leads to a sequence of indices $(i(s), j(s)) \in \{1, \ldots, n\} \times \{0, \ldots, k-1\}$ and elements $z^s \in \{0, \ldots, k-1\}^n$ so that

$$\hat{h}_\downarrow(\rho^{t-1}) - \hat{H}(0) = \langle \hat{w}^{t-1}, \rho^{t-1} \rangle = \sum_{s=1}^{n(k-1)} \rho_{i(s)j(s)} \big[ \hat{H}(z^s) - \hat{H}(z^{s-1}) \big],$$

where all $\rho_{i(s)j(s)}$ are arranged in non-increasing order. Because $\hat{h}_\downarrow$ and $\tilde{h}_\downarrow$ are defined as expectations of evaluations of $\hat{H}$ and $\tilde{H}$, they differ from at most $\eta$. We denote by $\tilde{w}^{t-1}$ the subgradient obtained from $\tilde{H}$.

We consider the iteration $\rho^t = \Pi_{\mathcal{S} \cap \mathcal{T} \cap [0,1]^{n \times (k-1)}}(\rho^{t-1} - \gamma \hat{w}^{t-1})$, where $\Pi_{\mathcal{S} \cap \mathcal{T} \cap [0,1]^{n \times (k-1)}}$ is the orthogonal projection on $\mathcal{S} \cap \mathcal{T} \cap [0,1]^{n \times (k-1)}$. From the usual subgradient convergence proof (see, e.g., [23]), we have:

$$
\begin{aligned}
\|\rho^t - \rho^*\|_F^2 &\leqslant \|\rho^{t-1} - \rho^*\|_F^2 - 2\gamma \langle \rho^{t-1} - \rho^*, \hat{w}^{t-1} \rangle + \gamma^2 \|\hat{w}^{t-1}\|_F^2 \\
&\leqslant \|\rho^{t-1} - \rho^*\|_F^2 - 2\gamma \langle \rho^{t-1} - \rho^*, \hat{w}^{t-1} \rangle + \gamma^2 B^2 \\
&= \|\rho^{t-1} - \rho^*\|_F^2 - 2\gamma \big[ \hat{h}_\downarrow(\rho^{t-1}) - \hat{H}(0) \big] + 2\gamma \langle \rho^*, \hat{w}^{t-1} \rangle + \gamma^2 B^2,
\end{aligned}
$$

using the bound $\|\hat{w}^{t-1}\|_F^2 \leqslant B^2 \leqslant nk\big[L_1/k + 2(nL_q/k)^q/q!\big]$. Moreover, we have

$$\langle \rho^*, \hat{w}^{t-1} \rangle = \langle \rho^*, \hat{w}^{t-1} - \tilde{w}^{t-1} \rangle + \tilde{h}_\downarrow(\rho^*) - \tilde{H}(0).$$

Since $\rho^* \in \{0,1\}^n$, there is a single element $s$ so that $\rho_{i(s)j(s)} - \rho_{i(s+1)j(s+1)}$ is different from zero, and thus $\langle \rho^*, \hat{w}^{t-1} - \tilde{w}^{t-1} \rangle$ is the difference between two function values of $\tilde{H}$ and $\hat{H}$. Thus overall, we get:

$$\|\rho^t - \rho^*\|_F^2 \leqslant \|\rho^{t-1} - \rho^*\|_F^2 - 2\gamma \big[ \tilde{h}_\downarrow(\rho^{s-1}) - \tilde{h}_\downarrow(\rho^*) - 2\eta \big] + \gamma^2 B^2,$$

which leads to the usual bound for the projected subgradient method, with an extra $2\eta$ factor, as if (up to the factor of 2) $\tilde{H}$ was submodular.

## E  Quadratic submodular functions

In this section, we consider the case where the function $H$ is a second-order polynomial, as described in Section 5.2 of the main paper.

Figure 4: Robust isotonic regression with decreasing constraints (observation in pink crosses, and results of isotonic regression with various losses in red, blue and black), from $25\%$ to $90\%$ of corrupted data. The dashed black line corresponds to majorization-minimization algorithm started from the observations.

Figure 5: Left: Effect of adding additional regularization ($n = 200$). Right: Non-separable problems ($n = 25$), distance to optimality in function values (and log-scale) for two discretization values ($k = 32$ and $k = 100$) and two orders of approximation ($q = 1$ and $q = 2$).

## E.1 Without isotonic constraints

We first consider the program without isotonic constraints, which is the convex program outlined in Section 5.2. Let $(Y, y)$ be a solution of the following minimization problem, where $\mathrm{diag}(A) = 0$ and $A \leqslant 0$:

$$\min_{Y,y} \frac{1}{2} \mathrm{tr}(A + \mathrm{Diag}(b))Y + c^\top y \quad \text{such that} \quad \forall i, \; Y_{ii} \leqslant y_i$$

$$\forall i \neq j, \; Y_{ij} \leqslant y_i, \; Y_{ij} \leqslant y_j,$$

$$\forall i \neq j, \; \begin{pmatrix} Y_{ii} & Y_{ij} & y_i \\ Y_{ij} & Y_{jj} & y_j \\ y_i & y_j & 1 \end{pmatrix} \succcurlyeq 0.$$

Some subcases are worth considering, showing that it is tight in these situations, that is, (a) the optimal values are the same as minimizing $H(x)$ and (b) one can recover an optimal $x \in [0,1]^n$ from a solution of the problem above:

– **"Totally" submodular**: if $c \leqslant 0$, then, following [18], if we take $\mathcal{Y}_{\mathrm{SDP}} = \{(Y, y), \forall i \neq j, Y_{ij}^2 \leqslant Y_{ii}Y_{jj}, \forall i, y_i^2 \leqslant Y_{ii} \leqslant 1, y_i \geqslant 0\}$, then by considering any minimizer $(Y, y) \in \mathcal{Y}_{\mathrm{SDP}}$ and taking $x = \mathrm{diag}(Y)^{1/2} \in [0,1]^n$ (point-wise square root), we have $H(x) = \frac{1}{2} b^\top \mathrm{diag}(Y) + \frac{1}{2} \sum_{i \neq j} A_{ij} Y_{ii}^{1/2} Y_{jj}^{1/2} + \sum_i c_i Y_{ii}^{1/2}$, and since $A_{ij} \leqslant 0$ and $c_i \leqslant 0$, it is less than $\mathrm{tr}\, Y(A + \mathrm{Diag}(b)) + c^\top y \leqslant \inf_{x \in [0,1]^n} H(x)$, and thus $x$ is a minimizer.

– **Combinatorial**: if $b \leqslant 0$, then we have: $H(x) = \sum_{i \neq j} A_{ij} x_i x_j + \frac{1}{2} \sum_{i=1}^n (-b_i) x_i (1 - x_i) + (c + b/2)^\top x$. Since $x_i x_j \leqslant \inf\{x_i, x_j\}$, $A_{ij} \leqslant 0$, $x_i(1 - x_i) \geqslant 0$ and $b_i \leqslant 0$, we have

$$\inf_{x \in [0,1]^n} H(x) \geqslant \inf_{x \in [0,1]^n} \sum_{i \neq j} A_{ij} \inf\{x_i, x_j\} + (c + b/2)^\top x.$$

Since the problem above is the Lovász extension of a submodular function the infimum may be restricted to $\{0,1\}^n$. Since for such $x$, $x_i x_j = \inf\{x_i, x_j\}$ and $x_i(1 - x_i) = 0$, this is the infimum of $H(x)$ on $\{0,1\}^n$, which is itself greater than (or equal) to the infimum on $[0,1]^n$. Thus, all infima are equal. Therefore, the usual linear programming relaxation, with $\mathcal{Y}_{\mathrm{LP}} = \{(Y, y), \forall i \neq j, Y_{ij} \leqslant \inf\{y_i, y_j\}, \forall i, Y_{ii} \leqslant y_i, 0 \leqslant y_i \leqslant 1\}$ is tight. We can get a candidate $x \in \{0,1\}^n$ by simple rounding.

– **Convex**: if $A + \mathrm{Diag}(b) \succcurlyeq 0$, we can use the relaxation $\{(Y, y), \; Y \succcurlyeq yy^\top, y \in [0,1]^n\}$ (which trivally leads to a solution with $x = y$). But we can also consider the relaxation $\mathcal{Y}_{\mathrm{cvx}} = \{(Y, y), \forall i \neq j, \begin{pmatrix} Y_{ii} - y_i^2 & Y_{ij} - y_i y_j \\ Y_{ij} - y_i y_j & Y_{jj} - y_j^2 \end{pmatrix}, \forall i, Y_{ii} \leqslant y_i\}$. We have then

$$\frac{1}{2} \mathrm{tr}(A + \mathrm{Diag}(b))Y + y^\top c \;=\; \frac{1}{2} \sum_{i \neq j} A_{ij}(Y_{ij} - y_i y_j)$$

$$+ \frac{1}{2} \sum_i b_i(Y_{ii} - y_i^2) + y^\top c + \frac{1}{2} y^\top (A + \mathrm{Diag}(b)) y$$

$$=\; \frac{1}{2} \sum_{i \neq j} A_{ij} \sqrt{Y_{ii} - y_i^2} \sqrt{Y_{jj} - y_j^2}$$

$$+ \frac{1}{2} \sum_i b_i(Y_{ii} - y_i^2) + y^\top c + \frac{1}{2} y^\top (A + \mathrm{Diag}(b)) y,$$

once we minimize with respect to $Y_{ij}$, from which we have, since $A_{ij} \leqslant 0$, $Y_{ij} = y_i y_j + \sqrt{Y_{ii} - y_i^2} \sqrt{Y_{jj} - y_j^2}$. If we denote $\sigma_i = \sqrt{Y_{ii} - y_i^2}$, we get an objective functiion equal to $\frac{1}{2} \sigma^\top (A + \mathrm{Diag}(b)) \sigma + \frac{1}{2} y^\top (A + \mathrm{Diag}(b)) y + c^\top y$, which is minimized when $\sigma = 0$ and thus $y$ is a minimizer of the original problem.

## E.2 Counter-example

By searching randomly among problems with $n = 3$, and obtaining solutions by looking at all $3^n = 3^3 = 27$ patterns for the $n$ variables being 0, 1 and in $(0, 1)$, for the following function:

$$H(x_1, x_2, x_3) = \frac{1}{200} \begin{pmatrix} x_1 \\ x_2 \\ x_3 \end{pmatrix}^\top \begin{pmatrix} -193 & -100 & -100 \\ -100 & 317 & -100 \\ -100 & -100 & -45 \end{pmatrix} \begin{pmatrix} x_1 \\ x_2 \\ x_3 \end{pmatrix} + \frac{1}{100} \begin{pmatrix} x_1 \\ x_2 \\ x_3 \end{pmatrix}^\top \begin{pmatrix} -146 \\ 136 \\ -216 \end{pmatrix},$$

the global optimum is $\begin{pmatrix} x_1 \\ x_2 \\ x_3 \end{pmatrix} \approx \begin{pmatrix} 1 \\ 0.7445 \\ 0 \end{pmatrix}$, the minimal value of $H$ is approximately $-0.3835$, while the optimal value of the semidefinite program is $-0.3862$. This thus provides a counter-example.

## E.3 With isotonic constraints

We consider the extra constraints: for all $(i, j) \in E$, $y_i \geqslant y_j$, $Y_{ii} \geqslant Y_{jj}$, $Y_{ij} \geqslant \max\{Y_{jj}, y_j - y_i + Y_{ii}\}$ and $Y_{ij} \leqslant \max\{Y_{ii}, y_i - y_j + Y_{jj}\}$, which corresponds to $x_i \geqslant x_j$, $x_i^2 \geqslant x_j^2$, $x_i x_j \geqslant x_j^2$, $x_i(1 - x_i) \leqslant x_i(1 - x_j)$, $x_i x_j \leqslant x_i^2$, and $x_i(1 - x_j) \geqslant x_j(1 - x_j)$.

In the three cases presented above, the presence of isotonic constraints leads to the following modifications:

– "Totally" submodular: because of the extra constraints $Y_{ii} \geqslant Y_{jj}$, for all $(i, j) \in E$, the potential solution $x = \mathrm{diag}(Y)^{1/2}$ satisfies the isotonic constraint and hence we get a global optimum.

– Combinatorial: nothing is changed, the solution is constrained to be in $\{0, 1\}^n$ with the extra isotonic constraint, implied by $y_i \leqslant y_j$, for all $(i, j) \in E$.

– Convex: the original problem is still a convex problem where the constraints $y_i \leqslant y_j$, for all $(i, j) \in E$, are sufficient to impose the isotonic constraints.