[Reviews · NeurIPS 2018]

Reviewer 1



This paper studies continuous submodular minimization under ordering constraints. The main motivation is to extend the settings in which efficient algorithms for isotonic regression exist. In particular, they extend such settings from convex objectives to submodular objectives. First, the authors show that this problem can be reformulated as a convex optimization problem with isotonic constraints. Since this convex problem is in a space of measures, it cannot be optimized exactly. Instead, the authors propose a discretization approach as in Bach18. The discretization approach is improved from the naive 1/eps^3 evaluations to 1/eps^2.33. Finally, the authors experimentally consider robustness to outliers with their approach. Examples of continuous submodular functions are listed, as well as a separate motivation for isotonic constraints, but it would have been nice if a concrete application of the problem studied was explained in detail. On the theoretical side, I don’t find the improvement from a naive 1/eps^3 number of evaluations to 1/eps^2.33 to be of major significance. I also found the paper to be very dense and often hard to follow sometimes. An example is that it is not clear how the noise and corruptions fit in the model. It would have been nice to have additional discussions about what corrupted data corresponds to in this setting.

Reviewer 2



In this paper the authors consider the problem of minimizing a continuous submodular function subject to ordering (isotonic) constraints. They first show that the problem can be solved if we first discretize it (per coordinate, not in [0,1]^n), and then solve the resulting discrete optimization problem using convex optimization. The fact that the problem is solvable in polynomial time is of course not surprising, because, as pointed out by the authors in lines 29-36, we can add a penalty to the objective that will implicitly enforce the constraints. However, this can significantly increase the Lipschitz constant of the objective, and that is why the authors take on an alternative approach. First, they prove that seen in the space of measures, the isotonic constraints correspond to dominating inequalities of the CDFs, which I guess is an intuitive result given the results known for the unconstrained case. For the discrete problems this adds another set of inequality constraints that have to be satisfied. Projection onto these constraints can be done using parametric maxflow among other techniques, so that the authors are able to achieve rates for this problem similar to those for the unconstrained one (sections 4.2 and 4.3). How this is exactly done is not clear, and I would suggest the authors to perhaps show how they reduce their problem to say parametric maxflow in 4.4, or at least in the supplementary. The authors later go on to discuss improved discretization algorithms. I would like to point out that all these schemes are uniform, in the sense that the points are equally spaced. What the authors analyze is the number of points that you need under different smoothness assumptions. Under uniform bounds on the gradients, they construct a surrogate for the function that is submodular and whose minimization results in faster rates. However, it is hard to evaluate as it requires the minimization of polynomials over box constraints - hence, I think of this algorithm to be of a more theoretical nature. Furthermore, the given guarantees, if I'm not mistaken, assume an exact evaluation of the surrogate. It is not clear if we instead solve approximate methods (e.g. SDP relaxations). Finally, I would like to remark that I did not read the proofs of the numerous convergence rates provided in the paper. Questions / Comments ---- 1. You project onto the constraints using parametric maxflow by solving a graph-cut problem with very large weights corresponding to E? How do you incorporate the box [0, 1] constraints? 2. Don't orthogonal projections and separable problems (section 4.4) reduce to the same problem? Can't you use one parametric flow also for the separable problems? If yes, I would suggest to present these together. 3. What is the purpose of section 5.2? I can not see how it is related to the discretization strategies. 4. l286 - Why only chain constraints? It is not clear from the Section 4.2, as there you add one constraint for each edge in E. 5. Is \Delta H in Fig.3. the difference of the computed optimum between two consecutive discretizations? Post-rebuttal: The rebuttal addressed all of my questions and comments. However, the more fundamental issues with the method that I have outlined (surrogate hard to evaluate, no guarantee under approximate surrogate evaluation) seem to hold, and that is why I will keep my score.

Reviewer 3



This paper studies continuous submodular minimization subject to ordering constraints. The main motivation is isotonic regression with a separable, nonconvex loss function. The starting point is the relationship between submodular minimization and convex minimization over a space of measures. It is demonstrated that ordering constraints correspond to stochastic dominance, and algorithms are proposed to enforce these constraints during optimization. The starting point is a simple way of discretizing the CDF corresponding to each variable. Then, improved discretization schemes are presented, depending on the availability of higher-order derivatives. Experimental results show that using a nonconvex loss function can improve robustness to corrupted data in isotonic regression. This paper is well-written, and makes a good set of technical contributions in showing how to incorporate ordering constraints and improve the discretization schemes. The experimental results nicely show the conditions under which the new techniques improve over standard methods. A couple questions: 1) Is there some intuition for why nonconvex losses like the proposed logarithmic function are more robust to outliers than squared or absolute value loss? 2) It makes sense that higher-order derivatives would improve the convergence rate in terms of number of iterations. However, this requires additional computation to get the Hessian (or even higher-order terms). Is the tradeoff worth it in terms of actual runtime?

Reviewer 4



This paper extends results of [1] on continuous submodular minimization by adding isotonic constraints to the objective function. The authors first apply the discretization procedure in [1] followed by projection onto the isotonic constraint set (e.g. max-flow), then propose to incorporate isotonic constraints into a refined discretization. Empirical results show increased robustness to outliers on synthetic data. Quality: The arguments appear to be correct, but the experimental evaluation is not very detailed. For example, only small synthetic examples with chain constraints are used, and the only comparison of naive vs improved discretization is Figure 3 (right) which does not verify the improved convergence rates. Moreover, the paper does not compare against previous work (separable convex isotonic regression [2]) where applicable. Clarity: While the paper reads well at a sentence level, it is often unclear where the exposition ends and the contributions begin. I think the paper would be much clearer if all of Sections 4-5 followed the format of Section 3, with Proposition/Theorem statements followed by proofs. Originality: While the combination of submodular minimization and isotonic constraints seems novel, this paper builds heavily on the results of [1]. The proofs of key propositions are less then 6 lines long, which suggests that the theory contribution is is a very straightforward extension. Significance: I am unsure of the impact of these results, given other recent work on constrained continuous submodular and DR-submodular minimization [3]. This is especially relevant because the current paper considers special cases of separable functions and quadratic functions. The authors have not clearly motivated the case for continuous submodular + isotonic constraints, and what advantages it has over other formulations. Question: How are isotonic constraints related to matroid constraints and/or down-closed convex constraints? Typos: Line 116 "Review of optimization of submodular optimization" Line 126 "When a single variables" Line 257 "dicretization" [1] Bach. Submodular functions: from discrete to continuous domains. Mathematical Programming, 2018. [2] Luss and Rosset. Generalized Isotonic Regression, 2014. [3] Staib and Jegelka. Robust Budget Allocation via Continuous Submodular Functions, 2017. ------------ UPDATE: While the author response addressed some of the reviewers' concerns regarding clarity, I agree with Reviewers 1 and 2 that the paper should be rejected

Reviewer 5



This paper considers the problem of continuous submodular minimization with box constraints and extra isotonic constraints, motivated by nonconvex isotonic regression. The paper gives an elegant algorithm for solving these constrained problems, by leveraging isotonic structure that is already present in the algorithms for continuous submodular minimization with box constraints from [1]. Beyond generalizing the approach of [1] to these constrained problems, the paper further shows that an extra smoothness condition on the objective yields better dependence the suboptimality criterion, for free, than thought before. Higher order smoothness can yield even better dependence. Finally, the paper shows the practical relevance of these techniques via a seemingly simple robust isotonic regression problem where natural convex formulations fail. A different (student t) model succeeds, but the objective is nonconvex; heuristics give poor performance, while the techniques from the paper solve the problem optimally and yield good performance on the regression task. Overall I think the paper has clean, well-executed theory and a motivating practical problem. Continuous submodular optimization is an important topic relevant to the NIPS community, as it presents a class of tractable nonconvex problems of use in machine learning applications (like nonconvex isotonic regression). Within this area of work, the paper is original and significant: while continuous submodular _maximization_ has seen much activity and many positive results in recent years, continuous submodular _minimization_ (this paper) seems more elusive. (beyond this paper I can only think of Bach '18 [1] and Staib and Jegelka '17, versus a multitude of maximization papers). This paper grants to us a new class of tractable constrained nonconvex problems. The paper is also clearly written and was a joy to read. I do have a few minor concerns/questions -- these are all practical considerations as I think the math is clean and not fluffy: - what is the wall-clock runtime like? - is it obvious that majorization-minimization is the most competitive alternative? (e.g. vs running projected gradient descent from random initialization) - I find the practical benefit of extra Taylor expansion terms in the approximation somewhat suspect -- it seems like you have to do a lot of extra work (both algorithmically and just by having to query q different derivative oracles) to get the improvement in epsilon dependence. It seems the fancy discretization was not really even used in the experiments (in "Non-separable problems" the authors compare the quality of approximations, but they seem to just use the usual discretization for the main regression experiments and just note that smooth objectives automatically get better approximation). Thus I am skeptical that "the new discretization algorithms are faster than the naive one." However the observation that smoothness implies better speed for free is still important. Overall, I think this is a solid paper with significant theoretical and algorithmic contribution, and it is also well-written. Some aspects of the experiments could be strengthened, but still I think this is a clear accept. minor comments: - the SDP relaxation is based on [15] which should be cited in the main text (it is cited in the appendix) - the point that \hat H is not technically submodular but we can still approximately solve probably should be stated more formally